Assessing the professional quality of life in the context of pediatric care

Alharbi Manal F. maalwahbi@ksu.edu.sa 1
Alkhamshi Aljawharah M. 2
1 Maternal & Child Health Nursing Department, College of Nursing, King Saud University , Riyadh , Saudi Arabia
2 Nursing Education Department, King Saud Medical City, Ministry of Health , Riyadh , Saudi Arabia
Thorrington Dominic
Electronic publication date: 2024 Mar 15
Publication date: 2024
Volume: 12
Electronic Location ID: e17120
Received 2023 Nov 17; Accepted 2024 Feb 26
Copyright: ©2024 Alharbi and Alkhamshi
Copyright year: 2024
Copyright holder: Alharbi and Alkhamshi
License: This is an open access article distributed under the terms of the Creative Commons Attribution License, which permits unrestricted use, distribution, reproduction and adaptation in any medium and for any purpose provided that it is properly attributed. For attribution, the original author(s), title, publication source (PeerJ) and either DOI or URL of the article must be cited.
License URL: https://creativecommons.org/licenses/by/4.0/

Keywords: Compassion, Fatigue, Burnout, Pediatric, Nurses

Funding: The Research Centre of the Female Scientific and Medical Colleges, Deanship of Scientific Research, King Saud University This research project was supported by a grant from the Research Centre of the Female Scientific and Medical Colleges, Deanship of Scientific Research, King Saud University. The funders had no role in study design, data collection and analysis, decision to publish, or preparation of the manuscript.

==============================
Background

This study examines the professional quality of life (ProQOL), including compassion satisfaction (CS) and compassion fatigue (CF), burnout (BO) and secondary traumatic stress (STS) among registered nurses in the context of pediatric care in Riyadh city in Saudi Arabia.

Methods

This study utilised a random multistage sampling method and a survey to collect data from 250 participants on personal characteristics, work-related aspects, lifestyle, and professional quality of life.

Results

The study participants were all female. Only 2.0% were under 25, with 27.2% aged between 25 and 30 years, 24.0% aged between 31 and 35, and 26.4% over 40 years old. The majority, 74%, were non-Saudis. The participants reported low compassion satisfaction (CS) and compassion fatigue (CF) levels. The CS, burnout (BO), and secondary traumatic stress (STS) scores were 3.23 ± 0.764, 2.75 ± 0.68, and 2.69 ± 0.54, respectively. The study found a significant difference in ProQOL level based on factors such as age, nationality, previous experience, department, shift length, overtime work, state of overtime (mandatory versus optional), and salary.

Conclusions

Working with vulnerable groups poses unique challenges for pediatric nurses. The nationality, salary, and state of overtime were found to be significantly different when compared to CS scores. Additionally, age, department, previous experience, shift length, and overtime work were very different compared to CF scores. However, lifestyle factors did not affect CS and CF scores significantly.

Implications

To improve their well-being and interest in activities, pediatric nurses should practice self-compassion and utilise available educational resources such as workshops. They should also strive to achieve a work-life balance and adopt healthy habits to reduce CF.

Introduction

The assessment of the professional quality of life in pediatric care can be challenging due to the unique needs of children. Pediatric nurses are crucial in providing quality healthcare to children (Alharbi & Alrwaitey, 2022). The care provided to different age groups, from infants to adolescents, requires professional management during distinct stages of motor, physical, emotional, and cognitive development (Alharbi & Alrwaitey, 2023). Nursing care in this context is challenging but essential for ensuring the best possible outcomes for young patients, especially when it comes to caring for sick children; it can be emotionally challenging to provide support to families who are experiencing emotional stress. According to a study by Hernández et al. (2019), many pediatric nurses have reported moderate-to-high levels of emotional exhaustion, depersonalisation, and low levels of personal accomplishment.

Pediatric nurses work in multidisciplinary teams as they may face ethical dilemmas when making decisions on behalf of the most vulnerable patient group or encountering child abuse or severe injuries (Larson et al., 2017). Pediatric nurses need clinical skills, compassion, patience, and an understanding of child development to navigate these challenges (Alharbi & Alrwaitey, 2023). Compassion is essential to healthcare, especially when treating young patients (Balakhdar & Alharbi, 2023). A scoping review conducted by Sinclair et al. (2020) has identified several factors associated with compassion in pediatric healthcare. These factors include continuity of care, effective communication, and care coordination (Sinclair et al., 2020). Developing relationships and delivering compassionate nursing care are critical when working with pediatric patients, and providing such care can enhance patients’ experiences and outcomes (Sinclair et al., 2020). It is widely recognised that the nursing profession places great importance on compassion, as it can significantly affect nurses’ physical and psychological well-being positively or negatively (Kase, Waldman & Weintraub, 2018). Exposure to recurring negative situations that require compassion, whether in a professional setting or a challenging relationship, can result in compassion fatigue (CF) (Craigie et al., 2016). According to Stamm (2010), CF is a mixture of physical, emotional, and spiritual exhaustion that arises from providing care for individuals who are experiencing intense emotional and physical pain. However, caregivers are not always emotionally distressed by exposure to such situations (Sacco & Copel, 2018). The term Compassion fatigue (CF) incorporates two comparable concepts: burnout (BO) and secondary traumatic stress (STS) (Peters, 2018). Burnout is characterised by a slow onset of emotional exhaustion, powerlessness, cynicism, and feelings of inadequacy related to one’s job (Hunt, Denieffe & Gooney, 2017). Secondary traumatic stress is a phenomenon that occurs when an individual becomes exposed to the traumatic experiences of others. Subsequently, they experience stress due to their desire to help the affected individual (Jenkins & Warren, 2012).

Nurses may experience emotional and physical exhaustion, restlessness, avoidance, anxiety, and fatigue as a result of BO and STS (Wolotira, 2023). BO and STS can lead to compassion fatigue and may affect the nurse’s ability to provide technically proficient and empathetic care, as per studies conducted by Shoji et al. (2015) and Sorenson et al. (2016). Simultaneously, working in a stressful environment, such as the nursing field, can expose the individual to BO and STS. Nurses face difficult working conditions every day. Nurses face various stressors that can cause work-related stress, leading to CF, stress negatively affects the quality of nursing work life of nurses (Alharbi et al., 2022). CF is a severe issue that can severely impact the nurses’ ability to perform their daily job activities (Upton, 2018). Healthcare facilities can benefit by focusing on measures to sustain a competent and caring nursing staff (Upton, 2018). A skilled and caring nursing staff is associated with patient satisfaction with nursing care and is considered a predictor of overall patient satisfaction with hospital care (Potter et al., 2010).

Nurses working in pediatrics are at risk of developing compassion fatigue due to secondary trauma (Berger et al., 2015) as the nurses deal with a population that requires significant sensitivity, which may increase the difficulty of their job. A similar study involving pediatric nurses in India indicated high stress in neonatal intensive care units (NICUs) (Amin et al., 2015). Another study conducted in the USA assessed CF among pediatric nurses and found lower CS and higher levels of BO, especially among those who had worked more than six years in pediatric nursing care (Berger et al., 2015), according to a study conducted in Saudi Arabia (SA) that included 46 nurses who participated in intensive care units, 2.75% suffered from BO syndrome (Habadi et al., 2018). A study in SA revealed very low CS, indicating an increased potential for the development of CF. Additionally, the study reported that coping strategies did not predict CS or CF, and resilience predicted CS and CF among nurses in critical care units (Alharbi, Jackson & Usher, 2020). A more recent study revealed moderate levels of CS, low levels of BO, and moderate levels of STS among pediatric nurses (Balakhdar & Alharbi, 2023).

Various factors influence pediatric nurses’ professional quality of life (ProQOL). The ProQOL of these nurses is affected by personal-related environmental, work-related environmental, pediatric-related environmental, and family-related environmental factors (Balakhdar & Alharbi, 2023). For instance, years of experience had a mixed effect on the level of CF and CS, according to a study by Kelly, Runge & Spencer (2015). The study surveyed 491 bedside nurses to explore predictors of CF and CS. Results indicated that as nurses gained work experience, they were more likely to have higher CF and lower CS.

In contrast, Mooney et al. (2017) conducted a study among oncology and intensive care unit (ICU) nurses to identify factors influencing CF and CS. They found that years of experience did not impact CS or BO levels but impacted CF, with scores decreasing with increasing years of experience. In Australia, a study by Jakimowicz, Perry & Lewis (2018) among ICU nurses with a bachelor’s degree examined determinants predicting and contributing to CS and CF. The findings showed that those with a bachelor’s degree had better CS scores than those with lower degrees (Jakimowicz, Perry & Lewis, 2018). According to a study by Helfrich et al. (2017), several work-related factors were associated with BO. The study investigated the relationships between primary care staff BO and the staffing and workload on their teams. The study included nurse care managers, clinical associates and administrative assistants. The overall level of BO was 41%. The strongest correlations in focused analyses were BO and having an entire team, turnover among co-workers and overcapacity of patient panels. The BO level was 30.1% lower for participants who worked on fully staffed teams with no turnover (Helfrich et al., 2017). Maintaining a balanced workload and a supportive team environment are key factors in reducing burnout and fostering a healthier work culture.

A study conducted by Beck, Cusson & Gable (2017) explored the support experience of NICU nurses after a traumatic event. The findings highlighted the importance of supportive colleagues, debriefing opportunities, prayer, speaking with a chaplain, and finding peaceful spaces for reflection. Conversely, ineffective communication, limited resources, unsupportive colleagues, and poor leadership were identified as factors that increased distress. Another study conducted by Barr (2017) in New South Wales, Australia, focused on 140 NICU nurses from four different NICUs. It examined the relationship between work stress, social support, compassion fatigue (CF), and compassion satisfaction (CS). The results revealed that workplace stress was associated with CF and CS. CF was predicted by role conflict and role overload, while role ambiguity was an individual predictor of CS. The presence of social support, when considering the influence of work stress, was found to be a significant predictor of both burnout and compassion satisfaction. However, in this study, it did not show any predictive power in relation to secondary traumatic stress.

Individualised social support based on intimacy predicted BO and CS, and individualised reassurance of worth predicted BO. Overall, both mediated and moderated social support positively affected CS and STS. CS did not influence the relationship between work stress and CF. Based on the findings, Barr (2017) concluded that role ambiguity, work overload, inadequate social support, and lack of appreciation were predictors within the NICU environment. In a study conducted by Wang et al. (2020) in China between 2018 and 2020, the impact of working and living conditions on nurses’ compassion, personal well-being, and secondary stress was examined. The study included 1,044 registered nurses from different agencies using the Professional Quality of Life Compassion Scale (ProQOL). The findings indicated that older married individuals with healthy sleep routines and sufficient sleep tended to have higher satisfaction with their relationships with others. Unsatisfied nurses exhibited personal behaviours, such as smoking, that negatively affected their compassion towards patients.

Nurses in Saudi Arabia face risks such as compassion fatigue (CF), burnout (BO), and secondary traumatic stress (STS). While research on CS and CF as a universal concept continues to expand, this study focuses specifically on CF and CS among registered pediatric nurses in SA. Previous studies have highlighted inconsistent findings and a need for more knowledge about the local context in SA. Understanding the factors contributing to this balance can help institutions retain nurses, reduce turnover, and create a better work environment. This, in turn, supports nurses in overcoming CF and enables employers to identify nurses experiencing CF. Assessing CF and CS among pediatric nurses can shed light on the emotional well-being of nurses in pediatric departments and provide a foundation for policymakers to improve work environments and enhance CS. The current study highlights CF and CS, highlighting pediatric nurses’ significant challenges. The results help fill the literature gap regarding the impact of CF and CS on pediatric nurses in SA. Following the conceptual model of ProQOL (Stamm, 2010), this study examines the differences in CS and CF among pediatric nurses in SA based on socio-demographic characteristics, work-related factors, and lifestyle patterns. Thus, the present study investigates CS and CF among pediatric nurses in Riyadh city in SA.

Materials & Methods

Study design

The study utilised a cross-sectional descriptive design. The target population included pediatric nurses working in critical and non-critical pediatric departments. The study occurred at Alymamh Hospital and King Saud Medical City (KSMC) Hospital. Alymamh Hospital specialised in maternal and child healthcare, while King Saud Medical City was biggher with a lot of medical buildings and facilities. Both hospitals had multiple pediatric departments, including critical care units, emergency room (ER) and special care units. They also had non-critical care units, including general surgical and medical wards. The total population of pediatric nurses in the two hospitals were 745 nurses. In this study, 250 pediatric nurses were included at Alymamh Hospital (n = 63, 25.2%) and King Saud Medical City (n = 187, 74.8%). All the included 250 nurses had completed the surveys. The study used a random multistage sampling method—a step-by-step approach to select participants. First, a random sample of clusters was chosen from two hospitals. Two groups were chosen from each hospital. In the second step, participants were selected within each set using systematic random sampling. The sample size for each cluster was determined based on the formula: (sample size/population size) × cluster size. As a result, the target population sizes for the groups were as follows: Alymamh Hospital - 154 nurses in critical units for cluster one, 38 nurses in general units for cluster two; KSMC Hospital - 317 nurses in critical units for cluster three, 236 nurses in available units for cluster four. Considering a 20% dropout rate, the final sample consisted of 250 pediatric nurses.

Instrument

The study included two parts. The first part covered personal characteristics, work-related aspects (like experience, salary, shift length, overtime arrangements, rota placement), and lifestyle aspects of nurses (such as hobbies, diet, sleep patterns, smoking, and social life). The second part was the ProQOL-5, a tool developed by Stamm in 2010. It comprised 30 items divided into three subscales (CS, BO, and STS). Each subscale had ten questions, scored from 5 (never) to 1 (very often), resulting in scores ranging from 10 to 50. However, items 1, 4, 15, 17, and 29 were scored opposite (from 5 = never to 1 = very often). The positive factor in ProQOL-5 was CS, while the negative factor was CF (BO and STS). Higher scores on the CS subscale indicated greater satisfaction, while higher scores on the BO subscale indicated higher risk. Higher STS subscale scores suggested that a nurse should review their feelings about work and the work environment and might need to discuss this with a supervisor, colleague, or healthcare professional. The validity and reliability of the ProQOL have been tested in different situations and languages. Geoffrion et al. (2019) examined the construct validity of the Professional Quality of Life (ProQoL) Scale, which measures psychological distress at work and the quality of work-life. Another study in Australia involving 1615 nurses found strong evidence of construct validity for the CS and CF scales (Sorenson et al., 2016). A survey by Balakhdar & Alharbi (2023) in Saudi Arabia demonstrated acceptable reliability of the ProQOL tool, with Cronbach’s alpha coefficients of 0.89 for CS, 0.83 for BO, and 0.78 for STS. The ProQOL-5 in this study showed acceptable internal reliability, with Cronbach’s α values of 0.90 for CS, 0.79 for BO, and 0.82 for STS.

Data collection procedure

The initial step was contacting the two hospitals and obtaining official approval for the study. Subsequently, data were collected using Google to create an online form. The form was installed on mobile drives like tablets, and it used a force-filling feature to ensure that participants filled in all the critical fields. The survey was shared with pediatric nurses through a hyperlink and an implied informed consent form by submitting the completed survey. This meant they agreed to participate in the study before proceeding to answer the questionnaire. Each participant could only use the link once and had the choice to select either the ”I agree” or “I do not agree” checkbox. Researchers visited the clinical area several days a week, met the participants and asked them to fill out the form on the device. At the same time, researchers waited nearby to help participants with any questions. Responses from participants were collected over six weeks from June 2021 to July 2021.

Data analysis

The Statistical Package for the Social Sciences (SPSS version 27; SPSS Inc., Chicago, IL, USA) was used for data analysis. A descriptive statistical analysis was conducted to determine the frequency and percentage of the variables in the study and to understand the distribution of the participants. Bivariate analysis, specifically one-way analysis of variance (ANOVA), was used to examine significant mean differences between CS, BO, STS, and other variables. This research considered a mean difference at the 0.05 level necessary. Additionally, the results indicated that the data exhibited a normal distribution based on histogram graphs. Following the ProQOL manual, the total scores were categorised as follows: ≤22 indicated a low level of ProQOL, 23–41 showed a moderate level, and ≥42 indicated a high level (Stamm, 2010).

Ethical considerations

This study was approved by the Institutional Review Board at King Saud University to conduct the study (H-01-R-053). The researchers followed all ethical standards of the 1964 Declaration of Helsinki and its later amendments or comparable ethical standards. Informed consent was obtained from all participants included in this study. The authors have permission from the copyright holders to use this instrument.

Results

Descriptive of participants, compassion satisfaction and compassion fatigue

This study included 250 pediatric nurses who worked at two designated hospitals. All the participants were females; only 2.0% were less than 25 years old, 27.2% were aged 25–30, 24.0% were 31–35, and 26.4% were more than 40. There were 74% non-Saudis. Approximately 46.9% had more than one child, and 10.7% had four or more children. A total of 81.2% of participants had three years or more experience in pediatric nursing, and 63.2% worked in critical care departments. A total of 77.2% had previous experience, 25.2% worked 8-hour shifts, and 34.8% worked overtime. Only 17.6% had a mandatory overtime schedule, and 48.4% had a fixed rota. Approximately 78.0% of the study participants had 5,000–9,000 SAR salaries. Only 15.2% had regular sleep patterns, and 38.4% had a healthy diet. Only 22.0% of the study participants socialised with their friends and family, 72.4% did not have a hobby, and only 2.0% smoked. The participants’ social, work, and lifestyle characteristics are presented in Table 1.

Table 1 The participants’ social, work and lifestyle characteristics (n = 250).

Socio-demographic factors	N	%	
Age	Less than 25 years old	5	02.0%	
25–30 years old	68	27.2%	
31–35 years old	60	24.0%	
36–40 years old	51	20.4%	
More than 40 years old	66	26.4%	
Nationality	Saudi	65	26.0%	
Non-Saudi	185	74.0%	
Social state	Married with kids	142	56.8%	
Married without kids	40	16.0%	
Not married with kids	8	03.2%	
Not married	60	24.0%	
Number of kids	1	36	22.5%	
2	75	46.9%	
3	31	19.4%	
4 or more	18	10.7%	
Experience	Less than 1 year	4	1.6%	
1–3 years	43	17.2%	
3 years or more	203	81.2%	
Department	Non-critical department	92	36.8%	
Critical department	158	63.2%	
Previous experience	Yes	193	77.2%	
No	57	22.8%	
Shift length	8 Hours	63	25.2%	
12 Hours	187	74.8%	
Working overtime	Yes	87	34.8%	
No	163	65.2%	
State of overtime	No. It is mandatory	44	17.6%	
Yes. It is optional	206	82.4%	
Rota changing	Yes	129	51.6%	
No	121	48.4%	
Salary	5,000–9,000 SAR	195	78.0%	
10,000–14,000 SAR	52	20.8%	
More than 14,000 SAR	3	1.2%	
Sleep Pattern	Regular	38	15.2%	
Irregular	212	84.8%	
Eating healthy diet	Yes	96	38.4%	
No	154	61.6%	
Frequency of socialising	Always	55	22.0%	
Often	86	34.4%	
Rarely	109	43.6%	
Practicing hobbies	Yes	69	27.6%	
No	181	72.4%	
Smoking	Yes	5	2.0%	
No	245	98.0%	
Sleep Pattern	Regular	38	15.2%	
Irregular	212	84.8%	

The total mean of CS among the study participants was 3.23 ± 0.76, which was relatively low. The actual mean for BO was 2.75 ± 0.68, which indicated a low level of BO among the study participants, and the total mean for STS was 2.69 ± 0.54, which showed a low level of STS. The prevalence of CS, BO and STS are presented in Fig. 1. Among the participants who took part in the study, CS was 17.20%, BO was 0.0%, and STS was 2.00%. Table 2 shows the mean and standard deviation of the CS, BO and STS scale items.

Relationship among study variables with ProQOL

The relationship between the participants’ characteristics, work-related aspects (like experience, salary, shift length, overtime arrangements, rota placement), lifestyle aspects, and ProQOL is presented in Table 3. There was a significant mean difference between the ages of participants and BO. There was a significant mean difference between nationality and CS. The participants’ departments showed a significant mean difference with BO. Also, participants’ previous experience, shift length and overtime work had a substantial relationship with STS. The state of overtime (mandatory versus optional) had a significant mean difference with CS. The participants’ salary levels had significant mean differences with CS only. Finally, there was no meaningful relationship between practising lifestyle factors and ProQOL Scale scores among the participants.

Discussion

The study added to our understanding of CS and CF by examining pediatric nurses from different backgrounds who work in two major hospitals in Riyadh. The results revealed that pediatric RNs had relatively low CS, BO, and STS levels. These findings contrast with a recent study conducted in Makkah, SA, which found that pediatric RNs had moderate levels of CS and STS (Balakhdar & Alharbi, 2023). However, our results align with the finding of low levels of burnout in the Balakhdar & Alharbi (2023) study, which indicates that pediatric Registered Nurses (RNs) may encounter similar levels of compassion fatigue CS, BO, and STS in various scenarios.

Figure 1 Prevalence of CS, BO and STS among the study participants.

Table 2 Descriptive of participants ProQOL Scale scores, compassion satisfaction and compassion fatigue (n = 250).

ProQOL Scale	Mean	SD	
Compassion Satisfaction	
I get satisfaction from being able to [help] people.	3.60	1.05	
I feel invigorated after working with those I [help].	2.95	1.12	
I like my work as a [helper].	3.25	1.10	
I am pleased with how I can keep up with [helping] techniques and protocols	3.18	1.07	
My work makes me feel satisfied.	3.24	1.14	
I have happy thoughts and feelings about those I [help] and how I could help them.	3.41	1.06	
I believe I can make a difference through my work.	3.23	1.05	
I am proud of what I can do to [help].	3.54	1.13	
I have thoughts that I am a ‘success’ as a [helper].	3.33	1.01	
I am happy that I chose to do this work.	3.50	1.10	
Total	3.23 ± 0.764	
Burnout	
I am happy.	2.77	.94	
I feel connected to others.	2.70	1.02	
I am not as productive at work because I am losing sleep over traumatic experiences of a person [help].	2.51	1.17	
I feel trapped by my job as a [helper].	2.57	1.24	
I have beliefs that sustain me.	2.88	1.14	
I am the person I always wanted to be.	2.76	1.12	
I feel worn out because of my work as a [helper].	2.94	1.08	
I feel overwhelmed because my case [work] load seems endless.	3.12	1.07	
I feel ‘bogged down’ by the system.	2.92	1.14	
I am a very caring person.	2.35	1.05	
Total	2.75 ± 0.68	
Secondary Traumatic Stress			
I am preoccupied with more than one person I [help].	3.13	1.01	
I jump or am startled by unexpected sounds.	2.75	1.04	
I find it difficult to separate my personal life from my life as a [helper].	2.57	1.10	
I think that I might have been affected by the traumatic stress of those I [help].	2.56	1.11	
Because of my [helping], I have felt ‘on edge’ about various things.	2.56	1.11	
I feel depressed because of the traumatic experiences of the people I [help].	2.70	1.05	
I feel as though I am experiencing the trauma of someone I have [helped].	2.58	1.00	
I avoid certain activities or situations because they remind me of the frightening experiences of the people I [help].	2.75	1.03	
As a result of my [helping], I have intrusive, frightening thoughts.	2.67	1.11	
I cannot recall important parts of my work with trauma victims.	2.63	.97	
Total	2.69 ± 0.54	

Table 3 Significant difference in the mean between the participants’ social, work and lifestyle characteristics and ProQOL Scale scores.

Factors	Professional quality life	
	CS	BO	STS	
	f	P	f	P	f	P	
Socio-demographic							
Age	1.970	.100	5.178	.001*	2.327	.057	
Nationality	15.139	.000*	16.120	.363	.037	.848	
Marital status	.379	.768	1.068	.363	1.843	.140	
Number of kids	1.724	.119	.899	.497	1.333	.246	
Experience	.143	.867	1.261	.184	.801	.766	
Work related		
Department	.224	.799	1.672	.024*	1.412	.082	
Previous experience	1.152	.318	1.062	.388	1.618	.026*	
Shift length	2.854	.060	1.174	.260	1.696	.016*	
Working overtime	.000	1.000	1.307	.151	1.913	.004*	
State of overtime	3.150	.045*	1.053	.399	1.176	.250	
Rota changing	1.670	.190	1.748	.016	1.179	.246	
Salary	4.046	.019*	.894	.621	.826	.731	
Lifestyle related		
Sleep pattern	.023	.977	.949	.554	3.769	.053	
Eating healthy Diet	1.755	.175	1.240	.182	.730	.851	
Frequency of socialising	1.949	.145	.973	.517	.902	.620	
Practicing hobbies	1.199	.303	1.481	.051	1.842	.176	
Smoking	.903	.407	1.150	.272	.924	.586	
Notes.

* Significant at p < 0.05 level.

The current findings in CS differ from similar studies conducted in other nursing specialities like ICU and ER. A survey by Varadarajan & Rani (2021) revealed a low level of CS among all ICU nurses, regardless of gender or work position. Likewise, a study conducted by Polat, Turan & Tan (2020) among nurses in various departments indicated a high level of CS across all departments. The results of the current study were quite similar to other studies involving pediatric nurses. Berger et al. (2015) conducted a study in the US with 239 participants and found that pediatric nurses generally experienced moderate to high levels of CS.

Additionally, Roney & Acri (2018) conducted a study which indicated that pediatric nurses had above-average levels of CS. Another survey by Polat, Turan & Tan (2020) among nurses in several departments showed a similar finding: nurses in all departments had a high level of CS. It appears to be an individual experience that varies among nurses, regardless of gender or department.

In the current study, CF was assessed by measuring the prevalence of BO and STS. Further analysis of the prevalence of BO among the study participants revealed that, on average, 82.20% of them experienced BO, but they all had a low level. This finding aligns with Varadarajan & Rani (2021), which found that male and female ICU nurses also had a low level of BO. Additionally, Roney & Acri (2018) found results similar to our study, showing that pediatric nurses had slightly lower than average levels of BO. However, the study by Berger et al. (2015) reported different findings, stating that 29% of pediatric nurses experienced high levels of BO. Another study by Akman et al. (2016) also reported a high level of BO among pediatric nurses. In terms of STS prevalence among the participants, 2.0% of the study sample experienced it. This finding is comparable to the study of Varadarajan & Rani (2021), where a group of male and female ICU nurses also had a low level of STS.

Similarly, Roney & Acri (2018) and our current study found that pediatric nurses had slightly lower than average levels of STS. However, Berger et al. (2015) reported different results, with 27% of pediatric nurses experiencing high levels of STS, which contrasts with our study, where only 2.00% had a high level of STS. Nurses’ compassion satisfaction and compassion fatigue vary based on their work settings. Intense work environments can increase CF, while less demanding work environments can boost CS. Additionally, a supportive work environment plays a significant role in promoting CS and reducing CF.

Certain personal and work-related aspects significantly differed with CS, BO, and STS among pediatric nurses. Results showed that the participants’ age was significantly related to BO, whereas there was no significant relationship between their age and their CS or STS. These findings were incongruent with Berger et al. (2015), who found that pediatric nurses older than 40 had a high level of CS, whereas nurses less than 40 had a lower CS and a higher level of BO. A study by Akman et al. (2016) aimed to determine BO factors among 165 pediatric nurses and showed different findings. It showed that marital status and older age were associated with a low level of BO. In the current study, marital status had no association with CS, BO or STS. In the present study, nationality was associated with CS, where Saudi nurses had a higher CS level than non-Saudis. In a similar survey of pediatric nurses by Berger et al. (2015), CS was significantly higher in Caucasians than in other races. The present study associated Certain work-related factors with CS, BO and STS. Nurses’ experiences had no significant relationship with CS, BO or STS. These findings were incongruent with Berger et al. (2015), in which pediatric nurses with 4–10 years of experience had a high level of BO, whereas nurses with more than 20 years of experience had a high level of CS and a low level of BO. Previous studies have demonstrated a relationship between socio-demographic characteristics, work and lifestyle factors on nurses’ susceptibility to CS and CF (Kelly, Runge & Spencer, 2015; Jakimowicz, Perry & Lewis, 2018; Mooney et al., 2017; Wang et al., 2020). In the current study, the pediatric nurses’ average CS differed from the findings of similar studies of other nursing specialities, such as ICU and ER. A survey by Varadarajan & Rani (2021) showed a low level of CS in all ICU nurses, regardless of their gender or work position.

Another interesting finding is that the participants’ department has a significant relationship with BO. Additionally, the participant’s previous experience, shift length, and working overtime had a substantial relationship with the STS. Regarding work-related factors, state of overtime (mandatory versus optional) and salary were only associated with nurses’ CS scores. However, this result was not similar to results reported in a study by Akman et al. (2016) in which the salary was associated with nurses’ BO levels. In the current study, no lifestyle factors were associated with CS, BO, or STS. This finding differed from Barr (2017), in which the social support of friends and family was positively correlated with nurses’ CS scores. This finding differed from Aytekin, Yilmaz & Kuguoglu (2013), who conducted a cross-sectional study involving nurses working in neonatal intensive care, which showed that poor quality of life correlated with BO. This difference across studies might be due to the cultural differences or differences in nurses’ values that influence their lives outside of work.

The current study was helpful as it contributed new evidence to the existing body of knowledge on the impact of CS and CF on registered pediatric nurses in SA. Understanding the factors that affect pediatric nurses’ professional quality of life can help healthcare organisations address CS and CF. This can be done by implementing interventions that enhance support systems, promote self-care, improve work environments, and provide ongoing training and education. These training areas may include stress management, emotional well-being, and self-care. Pediatric nurses can help themselves with CF. First, they should understand CF’s causes, symptoms, and habits to develop strategies for themselves and their colleagues. Second, they should practice self-compassion, which improves well-being and interest in activities. Third, they should use available educational resources like workshops to learn how to handle CF. Lastly, nurses should strive for a healthy work-life balance and incorporate positive habits into their daily routine. Various strategies have been suggested to help nurses cope with compassion fatigue, such as taking care of their mental and emotional well-being, maintaining physical health, nurturing spiritual needs, cultivating relationships with loved ones, focusing on personal behavior, and employing coping strategies specific to their work environment (Ondrejková, & Halamová, 2022).

Strengths and limitations

Using an online survey platform makes data collection faster and increases the response rate. Additionally, employing the probability sampling technique and implementing a step-by-step approach like random multistage sampling can help prevent bias. However, it is essential to note that there are limitations to the generalizability of these findings. The study utilised a descriptive cross-sectional design, which means it cannot establish cause-and-effect relationships or determine the direction of the relationship between variables. Even though the current study was conducted at two Ministry of Health hospitals, its external validity is limited since the results do not correspond to what is going on in other hospitals, including hospitals run by the security forces, hospitals run by the military, and hospitals run by the national guard.

Conclusions

Pediatric nurses face distinctive challenges when working with vulnerable groups. A relation was found between nurses’ nationalities, state of overtime (mandatory versus optional), and salaries and a higher CS. On the other hand, nurses’ ages and departments were related to a higher BO. Lifestyle factors were not found to impact CS or CF among nurses. However, work-related factors, such as previous experience, shift length, and working overtime, were found to have a significant contribution to STS. Nurses in pediatric areas can take proactive measures to manage CF effectively, such as understanding CF causes and symptoms, coping strategies through education, and prioritising their well-being by practising self-compassion to prevent psychosocial issues. In addition, they should take advantage of available educational resources such as workshops and seminars to enhance their knowledge of CF management. In these workshops and seminars, they will meet with other people to learn different techniques and strategies to understand and deal with their tiredness and stress. By attending these workshops and seminars, they will enhance their knowledge and abilities to take care of themselves and continue to provide care for others without becoming too exhausted.

Supplemental Information

Supplemental Information 1 Raw data

Supplemental Information 2 Questionnaire

Supplemental Information 3 STROBE checklist

Additional Information and Declarations

Competing Interests

Author Contributions

Human Ethics

Data Availability

The authors declare there are no competing interests.

Manal F. Alharbi conceived and designed the experiments, performed the experiments, analyzed the data, prepared figures and/or tables, authored or reviewed drafts of the article, and approved the final draft.

Aljawharah M. Alkhamshi conceived and designed the experiments, performed the experiments, analyzed the data, prepared figures and/or tables, and approved the final draft.

The following information was supplied relating to ethical approvals (i.e., approving body and any reference numbers):

This study was approved by the Institutional Review Board at King Saud University to conduct the study

The following information was supplied regarding data availability:

The raw data is available in the Supplemental File.

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
