# Peer review of "Assessing the professional quality of life in the context of pediatric care"

_PeerJ, doi:10.7717/peerj.17120_

## Round 0.1 · original submission · Minor Revisions

I'd like to bring your attention to the very detailed comments left by reviewer #1, in particular their calls for clarity on lines 90-91 and lines 94-99. You should certainly address their comments on Table 1 concerning the possible duplications of values in the table.

I think there is an error on one line in their review. "Line 265" should instead read "Line 254".

Reviewer #1 has kindly left several considerations for references but these are clarifications for your bibliography rather than calls to expand your bibliography with additional citations. These will help the reader check your citations following publication.

Reviewer 1 ·

Basic reporting

Thank you for the opportunity to review your manuscript. It was a pleasure to read, well written, and a well-executed study. You clearly stated the purpose and research gap. The thorough literature review included multiple studies from different countries. The manuscript demonstrated validity and reliability of the ProQOL, and the study design and instrument were clearly defined with demonstration of internal validity. The manuscript would be a great contribution to the literature. Line 348 is simply stated and objective, but as a reader of the manuscript, it provoked a strong sense of empowerment which will be excellent for those struggling with CF as CF can cause a sense of helplessness. I listed the following considerations regarding revisions:

-Noted inconsistency with the spelling of pediatric/paediatric throughout the manuscript. Consider using one spelling unless necessary to use both.
-To clarify meaning of state of overtime, instead consider “state of overtime (mandatory versus optional)” when first mentioned in the manuscript.
-Agree with the limitations and noted that there are several strengths (lines 355-357) mentioned in the limitations section. Consider changing the section title to “Strengths and Limitations”.
-Reducing depression is mentioned in the conclusion but the word depression is not mentioned much throughout the manuscript (although noted reference 6 includes depression). In view of this, consider a different word or phrase other than depression in the abstract/conclusion.
-Consider changing medical field in line 94 to perhaps nursing profession or paediatric nursing for enhanced relevancy to the topic of the manuscript.
-Line 134: The meaning of “participants who cared for a meeting within their capacity” is a bit unclear.
-I am unsure if it should read “lack of social support” as opposed to “social support” in line 145
-Consider briefly stating how is KSMC a more extensive medical complex (line 180) as the following two sentences describe the similarities between the hospitals.
-Line 265 I think should read 0.76 instead 76
-Line 257: “the majority of” is a bit unclear. Consider changing to something such as “a higher level of”.
-Consider mentioning Ridyadh (line 272) earlier in the manuscript, such as in the study design section. I would also consider including Saudi Arabia in the abstract background as per purpose statement in line 173.

There were a few references I could not locate within the manuscript (see below in “considerations for references” section for comments). In view of this, I included the following 3 considerations in case a reference was left out.
-Unsure if line 90-91, 94-99 should have references included
-Unsure if there should be a source cited in line 350 regarding depression
-Unsure if there should there be a year included after the American Nurses Association for a literature source in line 353

-I would love to read your ideas regarding examples of workshops and seminars for CF within the manuscript if able. This would be an excellent way for others to know where they can find help for themselves.
-The tables and figures are nicely done, but in second page of table 1, sleep pattern with N and % values are duplicated near middle and bottom of page. Also, if you are able to put table 3 all on one page, I think that would enhance readability of the table.


Considerations for references:
- Noted (AlharbiI et al., 2020b) is stated on line 113 but was unable to find Alharbi et al. (2020a) in the manuscript or reference list.
- Line 125: Jakimowicz et al. (2017) should be 2018 as per reference list.
- Geoffrion et al. (2019) is in the manuscript (line 207) but I could not locate it in the reference list.
- I was unable to locate the following references in the reference list within the manuscript: An et al. (2020), Chen (2022), Polit and Beck (2019).
- There are several titles of journals within the reference section that are not in italics
- Missing a period between MF (line 386)
- Reference 8: should the title of the article be first word capitalized and then the remainder of the title in lower case?
- If APA format, articles with 8 or more authors should be use first 6 authors then ellipsis then last author’s name (as done in reference 31)

Thank you again for the opportunity to review your manuscript.

Experimental design

No comments

Validity of the findings

No comments

Additional comments

No comments

·

Basic reporting

It's a good study. no comments.

Experimental design

No comments.

Validity of the findings

No comments.

Additional comments

The study is well-conducted and written.

---

## Round 0.2 · accepted · Accept

Thank you for the revisions to the manuscript; both reviewers are satisfied that the manuscript can now be published so I will recommend that your submission is accepted.

Reviewer 1 ·

Basic reporting

No comment

Experimental design

No comment

Validity of the findings

No comment

Additional comments

The study is well conducted and the manuscript is well written. I appreciate you taking the time to revise a bit of the manuscript from the suggestions. I thoroughly enjoyed reading the manuscript. It would be an excellent contribution to the literature. Thank you again for the opportunity to review your work.

·

Basic reporting

Now i feel that authors are done all necessary changes and the study looks well.

Experimental design

No comment.

Validity of the findings

No comments.

Additional comments

This stud looks well conducted and written now, so it can be accepted.